# Association of *BRAF* V600E Mutant Allele Proportion with the Dissemination Stage of Papillary Thyroid Cancer

**DOI:** 10.3390/biomedicines12030477

**Published:** 2024-02-21

**Authors:** Ivan Blazekovic, Ivan Samija, Josipa Perisa, Koraljka Gall Troselj, Tihana Regovic Dzombeta, Petra Radulovic, Matija Romic, Roko Granic, Ines Sisko Markos, Ana Frobe, Zvonko Kusic, Tomislav Jukic

**Affiliations:** 1Department of Oncology and Nuclear Medicine, University Hospital Center Sestre Milosrdnice, 10 000 Zagreb, Croatia; blazekovic90@gmail.com (I.B.); jperisa92@gmail.com (J.P.); romicmatija1@gmail.com (M.R.); roko.granic@gmail.com (R.G.); inessisko@gmail.com (I.S.M.); ana.frobe@kbcsm.hr (A.F.); tomislav.jukic@kbcsm.hr (T.J.); 2School of Medicine, Catholic University of Croatia, 10 000 Zagreb, Croatia; 3School of Dental Medicine, University of Zagreb, 10 000 Zagreb, Croatia; 4Laboratory for Epigenomics, Division of Molecular Medicine, Ruđer Bošković Institute, 10 000 Zagreb, Croatia; troselj@irb.hr; 5Department of Pathology Ljudevit Jurak, University Hospital Center Sestre Milosrdnice, 10 000 Zagreb, Croatia; tihana.dzombeta@mef.hr (T.R.D.); pradulovi@gmail.com (P.R.); 6School of Medicine, University of Zagreb, 10000 Zagreb, Croatia; 7Croatian Academy of Science and Arts, 10000 Zagreb, Croatia; zaklada@hazu.hr; 8Faculty of Medicine Osijek, Josip Juraj Strossmayer University of Osijek, 31 000 Osijek, Croatia

**Keywords:** papillary thyroid cancer, dissemination, *BRAF* V600E, tumor heterogeneity, qPCR

## Abstract

The early identification of aggressive forms of cancer is of high importance in treating papillary thyroid cancer (PTC). Disease dissemination is a major factor influencing patient survival. Mutation status of *BRAF* oncogene, *BRAF* V600E, is proposed to be an indicator of disease recurrence; however, its influence on PTC dissemination has not been deciphered. This study aimed to explore the association of the frequency of *BRAF* V600E alleles in PTC with disease dissemination. In this study, 173 PTC samples were analyzed, measuring the proportion of *BRAF* V600E alleles by qPCR, which was then normalized against the proportion of tumor cells. Semiquantitative analysis of BRAF V600E mutant protein was performed by immunohistochemistry. The *BRAF* V600E mutation was present in 60% of samples, while the normalized frequency of mutated *BRAF* alleles ranged from 1.55% to 92.06%. There was no significant association between the presence and/or proportion of the *BRAF* V600E mutation with the degree of PTC dissemination. However, the presence of the *BRAF* mutation was significantly linked with angioinvasion. This study’s results suggest that there is a heterogeneous distribution of the *BRAF* mutation and the presence of oligoclonal forms of PTC. It is likely that the *BRAF* mutation alone does not significantly contribute to PTC aggressiveness.

## 1. Introduction

Thyroid cancer is a common tumor with a significant increase in incidence. Croatia ranks 4th according to the age-standardized incidence rate in Europe [1,2]. In recent years, increased screening in the general population, primarily in developed countries, has led to an increase in the incidence of PTC [3,4]. According to the data from the Surveillance, Epidemiology, and End Results-9 (SEER-9) cancer registry program [5], the mortality rate for thyroid cancer is increasing as well. This suggests that the increase in incidence cannot solely be attributed to improvements in diagnostic methods, but also to changes in disease aggressiveness, likely caused by environmental factors [6,7].

Although considered to be an indolent tumor, that most often occurs as a localized disease or with lymphogenic metastases, PTC can still initially present with hematogenous metastases in approximately 4% of patients, leading to an increased risk of recurrence and an increased mortality rate [8,9,10]. It is crucial to differentiate the indolent type of PTC from the more aggressive forms as early as possible. The discovery of new subclinical cases and the treatment with long-term follow-up of patients with non-aggressive diseases represent a significant burden for the patient, as well as for the healthcare system. The discovery of molecular markers that would allow for early recognition of well-differentiated thyroid tumors with a higher potential of disease dissemination, would enable timely recognition of high-risk patients. Research on genetic changes, especially mutations that may affect the biological behavior of tumors, will allow for a better understanding of the fundamental molecular features that determine the aggressiveness of thyroid cancer [11,12].

Well-differentiated thyroid tumors, according to the standard accepted model of oncogenesis, arise from the gradual malignant transformation of thyroid cells [13]. The most common mutations in PTC affect *BRAF* (B-Raf proto-oncogene serine/threonine-protein kinase) and *RAS* genes. These mutations, as well as RET/PTC rearrangements, are considered to be driving mutations. There are also *TERT* (Telomerase Reverse Transcriptase) gene mutations that occur at a later stage of tumor development and additionally contribute to tumor aggressiveness [12,14,15]. Mutations in the aforementioned genes are present in approximately 70% of all thyroid cancers [16]. Recent data suggest that certain molecular profiles, such as the coexistence of *BRAF* with other oncogenic mutations, TERT promoter mutations, or TP53 mutations may be more specific markers of a less favorable outcome of PTC [17]. Diagnostic and predictive value of *RAS* mutation are limited in thyroid nodular disease [18]. However, the presence of *BRAF* or *RAS* mutations enhances the prognostic effects of *TERT* promoter mutations [19]. It has also been shown that *TERT* promoter mutations significantly increase the risk of both recurrence and mortality in high-risk (according to American Thyroid Association, ATA) and TNM stage III/IV groups [19]. Currently, ATA guidelines suggest high risk of recurrence in patients with both *BRAF* and *TERT* mutation. On the other hand, the presence of *BRAF* mutation is only associated with a low and intermediate risk of recurrence. Clinical significance of isolated *BRAF* mutation for risk stratification is not clear, and it is not routinely recommended in postoperative care of PTC [8].

Mutation *BRAF* V600E is the most common mutation in PTC, with an average incidence rate of approximately 45%, ranging from 27.3 to 87.1% [20,21]. This type of mutation has been discovered in more than 90% of PTCs with known *BRAF* mutations. If BRAF V600E is present, the activity of BRAF kinase is 480 times higher than in the BRAF wild-type cells [22]. The mutation is in the activating domain of the protein (CR3), which is composed of amino acids 457–717 [23]. The mutation itself results in constitutive activation of the MAP-kinase signaling pathway, which can result in aberrant gene expression, accelerated proliferation, and avoidance of apoptosis. Although extensively investigated, the clear association between the presence of the *BRAF* V600E mutation and clinicopathological prognostic factors of PTC has not yet been demonstrated [24]. Accordingly, mutation in the *BRAF* gene, as a prognostic factor in PTC, is still questionable [25,26]. Recent research has revealed the molecular mechanisms determining the progression and aggressiveness of PTC due to *BRAF* mutation [27,28]. These mechanisms include the down-regulation of major tumor suppressor genes and thyroid iodide-metabolizing genes, as well as the up-regulation of molecules that promote cancer, such as vascular endothelial growth factor (VEGF) A and its receptor VEGF receptor (VEGFR)-2, the main regulators of the angiogenic process [28].

The results of research based on the association of the mutation with the clinicopathological prognostic factors of PTC (bilaterality, multifocality, thyroid capsule infiltration, and lymph node metastases) remain controversial. While some studies show a connection between the presence of a mutation and the aforementioned factors [29,30], the results of other studies did not demonstrate such connection [26]. A study showed that presence of the *BRAF* mutation was significantly associated with increased cancer-related mortality in PTC patients [31]. The study found that the recurrence rates were higher in *BRAF* mutation-positive patients (47.71 per 1000 person-years) compared to those who were *BRAF* mutation-negative (26.03 per 1000 person-years). This difference was statistically significant (*p* < 0.001), with a hazard ratio (HR) of 1.82 (95% CI, 1.46 to 2.28). On the other hand, another study found that this difference remained significant even after adjusting for clinicopathogenic factors in a multivariable model [32]. In all studies cited, only the presence or absence of the *BRAF* V600E mutation was explored, without a further quantification of mutated alleles, in positive cases. However, *BRAF* mutation may only be present in a small proportion of cells. This is especially important when a large lymphocytic infiltrate is present, as in the case of Hashimoto’s thyroiditis, where mutated, tumor-originating *BRAF* alleles become “diluted” by wild-type alleles originating from infiltrating lymphocytes [33]. Furthermore, the monoclonality of PTC has recently been questioned by multiple studies [34,35,36,37]. These studies claim that there is a heterogeneous distribution of cells containing mutation *BRAF* V600E, within the tumor [23,24,25,26]. The heterogenous distribution could be one of the proposed explanations for the unclear relationship between the presence of the *BRAF* V600E mutation and the prognostic factors that determine the aggressiveness of PTC [34,35,36,37]. It has been also suggested that *BRAF* V600E mutation is heterogeneously distributed and limited to part of the tumor cells in most PTC samples [35,38]. The aim of this research is to study the association of the proportion of tumor cells with *BRAF* V600E coding allele and the dissemination of the disease.

## 2. Materials and Methods

### 2.1. Participants

We retrospectively collected data from 173 adult patients diagnosed with PTC at the Department of Oncology and Nuclear Medicine of the University Hospital Center Sestre milosrdnice, Zagreb, Croatia. All procedures were in accordance with the 1964 Helsinki Declaration and its later amendments, or comparable ethical standards. The study was approved by the institutional Ethics Committee (University Hospital Center Sestre milosrdnice; approval No. EP-9941/19-7), and informed consent was obtained from all the patients. Paraffin blocks of the primary thyroid tumor after total thyroidectomy were collected. We also assessed data on the pathohistological (PH) characteristics of the tumor, clinical data, and data related to postoperative oncological treatment.

Based on the PH findings, the following data were collected: size of the primary tumor, area of thyroid involvement (left or right lobe, isthmus), dissemination within the thyroid gland, infiltration of the thyroid capsule, extrathyroidal expansion, and angioinvasion (i.e., tumor cells in vascular spaces determined by histopathologic examination).

Based on the findings of postoperative oncological treatment, as well as the findings of PH, it was established whether there were metastatic positive lymph nodes in the neck and distant metastases sites. With respect to positive lymph nodes, the largest metastatic lymph node was analyzed, and breakthrough of the lymph node capsule, if present, was noted.

### 2.2. Histological Analysis of the Primary Tumor

After fixation in 10% buffered formalin and dehydration in increasing concentrations of alcohol, the removed thyroid tissue was embedded in paraffin blocks. Paraffin sections were cut at 5 μm. Deparaffinization in xylene and hemalaun and eosin staining (H&E) was performed by the standard method. Sections were cut from each block in the following manner: the first section stained with H&E was used to assess the proportion of tumor cells expressed as a percentage in relation to all remaining cells in that section. The next few sections were then used for the extraction of genomic DNA for *BRAF* V600E mutation analysis. An additional section obtained from 45 selected paraffin blocks (positive and negative in relation to the presence of the mutation *BRAF* V600E) was used for analyzing BRAF on a protein level, by immunohistochemistry.

### 2.3. DNA Extraction and BRAF V600E Mutation Detection

DNA was extracted following the manufacturer’s instructions using the QIAamp DNA FFPE Tissue kit (Qiagen, Hilden, Germany). The concentration and quality of the extracted DNA was determined using NanoDrop 2000 (Thermo Fisher Scientific, Waltham, MA, USA) spectrophotometer. Only DNA of OD (optical density coefficient, 260/280) 1.70 ± 20% was used for further experiments.

For mutation analysis, the target DNA was amplified and detected using TaqMan^®^ Mutation Detection Assay (Applied Biosystems, Foster City, CA, USA) on 7500 Real-time PCR System (Applied Biosystems, Foster City, CA, USA) following the manufacturer’s instructions. Each reaction was performed in triplicate, using 25 ng of DNA. In each reaction, a negative and a positive control were used.

Analysis of qPCR reaction data was performed with Mutation Detector software package (Applied Biosystems, Foster City, CA, USA). From the Ct values obtained with the TaqMan mutation detection probe and the TaqMan reference gene probe, the normalized ΔCt value was calculated. The percentage of mutated alleles was calculated from the normalized ΔCt according to the following Formula (1):%mutation = [1/2ΔCt ÷ (1/2ΔCt + 1)] × 100%(1)

Since the assessment of genetic heterogeneity in tumor samples can be imprecise due to the presence of non-tumor cells, the data related to the percentage of mutation were normalized with respect to the previously determined percentage of tumor cells in the sample, according to the following Formula (2) as suggested by de Biase et al. [36]:Normalized % mutation in tumor =% mutation/% tumor cells in the section of the preparation(2)

### 2.4. Immunohistochemical (IHC) Analysis

The presence of *BRAF* V600E mutation (Ventana anti-*BRAF* V600E/VE1, Roche, Mannheim, Germany) was determined in 45 tumor sections using the BenchMark GX automatic multifunctional IHC staining system (Roche, Basel, Switzerland) with the OptiView DAB Detection Kit (Ventana Medical Systems Inc., Oro Valley, AZ, USA), following the manufacturer’s instructions. The antibody used was previously validated for detecting BRAF V600E protein in thyroid cancer [39]. Post-counterstaining was performed using Bluing Reagent, after Hematoxylin counterstaining. Metastatic melanoma tissue with a confirmed *BRAF* gene mutation was used as a positive control. In the case of a positive finding, the percentage of PTC cells with expressed altered protein in the analyzed sample was estimated.

### 2.5. Statistical Methods in Data Processing

The comparison of the clinicopathological features between the groups with and without *BRAF* V600E mutation was performed using the χ^2^ test, Mann–Whitney, and *t*-test, while the association between the frequency of *BRAF* V600E mutated alleles and the extent of PTC was tested with a one-way ANOVA and Spearman test. A comparison of the concordance between the *BRAF* mutation results and the results obtained by immunohistochemistry was tested using Cohen’s kappa method. Statistical analysis was performed using the Statistica program (StatSoft, Inc., Hamburg, Germany). A statistically significant result was considered to be *p* < 0.05.

## 3. Results

### 3.1. Association between BRAF Mutation with Disease Dissemination and Characteristics of the Disease

Patient and tumor characteristics are presented in Table 1. We analyzed the association between the existence of the *BRAF* V600E mutation as well as the normalized values of the proportion of the mutated allele *BRAF* V600E with disease dissemination, age, sex, tumor size, capsule breakthrough, angioinvasion, dissemination within the thyroid gland, extrathyroidal expansion and infiltration of the lymph node capsule, in case of local metastases.

There was no significant association between the existence of the *BRAF* V600E mutation and disease dissemination (χ^2^ (1, N = 173) = 3.737; *p* = 0.154). The results of our study show a lack of association of the *BRAF* mutation status with gender (χ^2^ = 1.2859; *p* = 0.257), tumor size (Mann–Whitney U test; U = 2890.5, Z = −1.69, *p* = 0.09), extrathyroidal expansion (χ^2^ (1, N = 164) = 0.028; *p* = 0.866), capsule infiltration (χ^2^ (1, N = 167) = 0.373; *p* = 0.541), or dissemination within the thyroid gland (χ^2^ (1, N = 169) = 0.614; *p* = 0.433). There was a statistically significant association between the presence of *BRAF* V600E and angioinvasion (χ^2^ (1, N = 132) = 4.112; *p* = 0.043). Also, age at diagnosis was significantly higher in the group with *BRAF* V600E mutation (Mann–Whitney U test; U = 2613.5, Z = 3.02, *p* = 0.03). If lymphatic metastases were present, the relationship between the lymph node capsule infiltration and the existence of the *BRAF* V600E mutation was analyzed, but no statistically significant association was found (χ^2^ (1, N = 47) = 1.035; *p* = 0.309).

Further analysis of the association of the normalized frequency of mutated allele *BRAF* with disease dissemination, presented in Table 2, did not show a statistically significant association (ANOVA test, F(2, 97)=0.28287, *p* = 0.754). No statistically significant association of heterogeneity with dissemination was shown, even when patients were divided into those with a frequency of tumor cells below 40%, 40 to 60%, and those above 60% (Table 3; χ^2^ (4, N = 100) = 3.257; *p* = 0.515). There was also no statistically significant association with thyroid capsule infiltration (Mann–Whitney U test; U = 808, Z = 0, *p* = 1.000), tumor angioinvasion (Mann–Whitney U test; U = 72, Z = −1.663, *p* = 0.096), dissemination in the thyroid gland (Mann–Whitney U test; U = 1052, Z = 0.557, *p* = 0.576), and extrathyroidal expansion (Mann–Whitney U test, U = 776, Z = 0.54, *p* = 0.589). Although the age at the time of diagnosis was significantly higher in the group with *BRAF* mutation, further analysis of the group with *BRAF* mutation showed no statistically significant correlation between the percentage of mutated allele and age at diagnosis (Spearman’s test, N = 100; t = 0.311; *p* = 0.756). There was also no statistically significant correlation between the percentage of mutated allele and tumor size (Spearman’s test, N = 98, R = 0.0196; t = 0.192; *p* = 0.847).

### 3.2. Heterogeneity of Mutated BRAF V600E Allele (qPCR)

The mutation *BRAF* V600E in primary tumors was detected in 104 (60.1%) of a total of 173 tumors. The percentage of *BRAF* V600E alleles assessed with qPCR ranged from 0.5% to 41.6%, with a median of 5.91%. The estimated proportion of tumor cells in the sample ranged from 5% to 90% with a median of 30%. The normalized proportion of mutated *BRAF* V600E allele ranged from 1.55% to 92.06%, with a median of 23.33% (Figure 1). We observed a wide range of values of the normalized proportion but noticed that the majority of tumors (76%) had less than 40% mutated *BRAF* alleles (Table 3).

### 3.3. Comparison of qPCR and Immunohistochemistry Method

Staining of BRAF V600E mutated protein by IHC (Figure 2) was performed on 45 samples, 43 of which were included in further analysis. The two remaining samples were not further analyzed due to unspecific cytoplasmic staining of tumor cells and staining of stromal cells. Of 43 samples, 13 were negative for the presence of BRAF-mutated protein. The same samples were also negative when analyzed by qPCR. Of a total of 30 IHC-positive samples, 29 were positive by qPCR analysis. The only sample that was negative for the presence of the *BRAF* V600E by qPCR and positive by immunohistochemical staining had a percentage of BRAF mutated protein-containing cells of only 5%. Cohen’s kappa test showed a very good agreement between the results of BRAF mutation analysis obtained by two methods (Cohen’s Kappa = 0.9460; % of agreement: 97.67).

## 4. Discussion

Due to the fact that the mutation *BRAF* V600E presents the most common mutation in PTC, it is not surprising that numerous studies are exploring the association between *BRAF* mutation and prognostic factors of the disease. So far, published studies have shown inconsistent results, and further analyses, based on the association of allele mutation frequency and tumor characteristics, are needed. In our PTC study, the proportion of *BRAF* V600E positive samples was 60%, which is within the expected, previously described range of approximately 45% [21,27]. We analyzed the association between the existence of mutation *BRAF* V600E and the normalized values of mutated alleles with all available PTC features. Our results show no statistically significant association between the existence of mutation *BRAF* V600E and metastatic spreading of PTCs (*p* = 0.154) nor pathohistological features of the primary tumor, except for tumor angioinvasion (*p* = 0.043). There was no significant difference in gender distribution between the groups. Still, the *BRAF* V600E-positive patients were significantly older at the time of diagnosis (*p* = 0.003), although there was no statistically significant correlation between the percentage of mutated allele and the age at diagnosis (*p* = 0.756). In several previous studies, a statistically significant association between the presence of the *BRAF* V600E mutation and distant PTC metastases was described [40,41]. On the other hand, the impact of *BRAF* V600E mutation on lymphogenic local metastases has been investigated in several studies with contradictory results. Some of them describe a statistically significant association between the presence of *BRAF* V600E mutation and the existence of local metastases [40,42,43,44]. However, a study performed by Sapio et al. reports a statistically significant association of mutation *BRAF* V600E with the absence of local metastases (χ^2^, N = 43; *p* = 0.011) [45]. Further on, a study conducted by Gandolfi and colleagues did not find an association of *BRAF* mutation with either local or distant metastases [37]. In their study, the mutation *BRAF* V600E was found in 18 of 37 PTCs without metastasis (48.6%) and in 40 of 95 PTCs with local metastases (42.1%). The mutation was also detected in 33.3% of PTCs with distant metastases [37]. The results reported by Kim et al. and Finkel et al. present a higher percentage of mutated alleles in patients with local metastases [46,47]. Kim found that lymph node metastases were more common in PTCs with a high abundance (≥20%) of mutant alleles than in those with a low abundance of mutant alleles (*p* = 0.010) [47]. Although we noticed a trend of higher frequency of *BRAF* V600E alleles in patients with distant metastases, compared to patients without metastases as well as to patients with local metastases, there was no statistical significance (*p* = 0.515). To further examine the clinical importance of the *BRAF* gene mutation, Cheng et al. investigated the association between the frequency of mutated alleles and clinicopathological features and showed a significantly higher frequency of mutated alleles in larger tumors (p = 0.0001) and tumors with infiltration of the thyroid capsule (*p* = 0.034) [38]. Our results show no such association between the frequency of mutated alleles and tumor size (*p* = 0.847), thyroid capsule infiltration (*p* = 1.000), tumor angioinvasion (*p* = 0.096), dissemination in the thyroid gland (*p* = 0.576), or extrathyroidal expansion (*p* = 0.589).

According to our results, there is a heterogeneous distribution of *BRAF*-mutated alleles in the PTC tissue. Normalized frequency values of the *BRAF* gene allele ranged from 1.55% to 92.06%, with a median of 23.33%, indicating oligoclonal development. The oligoclonal pattern of BRAF mutation presence strongly indicates a heterogenous cellular composition of the PTC, at the time of surgical extirpation. This, of course, does not exclude monoclonal origin of the tumor. These results are in accordance with the results of Guerra and colleagues who described the oligoclonal existence of the mutation in most of the analyzed PTC samples [35]. They demonstrated that only in 4/41 PTC samples did mutation *BRAF* V600E range from 44.7 to 43.7%, and in 27/41 samples ranged from 25 to 5.1%. This means that 66% of PTCs with *BRAF* mutation had the frequency of the mutation alleles less than 25%, suggesting that *BRAF* mutation is often an oligoclonal event [35]. Similar results were reported by Kim et al., who found that the median frequency of the *BRAF* V600E mutated allele was 20%. Similarly so, Finkel et al. described the range of *BRAF* mutated allele frequency from 12% to 39% [46,47]. Research by Gandolfi et al. and De Biase et al. also showed the presence of heterogeneity of *BRAF* mutation and confirmed the results obtained by pyrosequencing, allele-specific locked nucleic acid PCR, and next-generation sequencing by immunohistochemical analysis of the mutated BRAF protein [36,37]. Gandolfi et al. demonstrated that BRAF mutation is frequently an oligoclonal event, present in 58.6% (34 out of 58) of PTCs, with a mutation percentage ranging between 20% and 35% [37]. On the other hand, not all studies are showing results in line with PTC heterogeneity.

Colombo et al. showed the existence of *BRAF* mutation heterogeneity only in a small number of cases (10/87) presenting a mean normalized allelic frequencies of 49.91 ± 11.22 (mode: 51, range 22–100) for mutation *BRAF* V600E [25]. In our study, the heterogeneity was clearly presented by the results of qPCR, but the number of positive *BRAF* V600E cells determined by IHC seemed higher than one would expect based on qPCR analysis. In other words, the results obtained on the protein level did not support the concept of heterogeneity and further suggested a homogeneous distribution of the mutation, which is in accordance with the results of Ghosein and colleagues [48].

### Limitations

The research was performed on consecutive slides of the paraffin block, obtained from the area with the largest diameter of the tumor. Although the slides were only 5 µm-thick, there is still a possibility that the section(s) used for DNA isolation differed with respect to the proportion of tumor cells which were estimated in the first slide cut off. 

However, the average diameter range of the nucleus of PTC is up to 5.92 µm [49] or in the range of 4.7–9.6 µm [50]. The next possible drawbacks are the decision on the optimal section of the tumor for the H&E staining, and presence of DNA degradation. The assumption is that the site of the largest diameter of the primary tumor is the best for analysis. This is where we expect the most tumor cells and a representative sample for detecting mutations of interest. However, if there is a heterogenous distribution of *BRAF* mutated allele in PTC tissue, a late mutation can occur in any part of the tumor, including its periphery.

## 5. Conclusions

This study demonstrates that neither the presence of the *BRAF* V600E mutation nor the proportion of cells with a mutated allele is significantly linked to the aggressiveness of PTC. These findings suggest that the role of the *BRAF* V600E mutation as a negative prognostic marker in PTCs needs to be reevaluated. According to the data available, heterogeneity of mutations in PTC are not as common as in other types of cancer such as melanoma, but it still exists. Thus, the potential multifocal growth of tumors with mutation *BRAF* V600E should be considered. However, even the demonstrated heterogeneity is not sufficient for explaining the ambiguous role of *BRAF* V600E mutation as a prognostic marker which may have a potential to affect important clinical decision. It seems that it still represents a hardly understandable influence on the clinical and pathological presentation of PTC. Furthermore, as expected, the proportion of patients with a mutation increases with age, and this factor should also be considered when evaluating clinical significance of the *BRAF* V600E mutation.

## Figures and Tables

**Figure 1 biomedicines-12-00477-f001:**
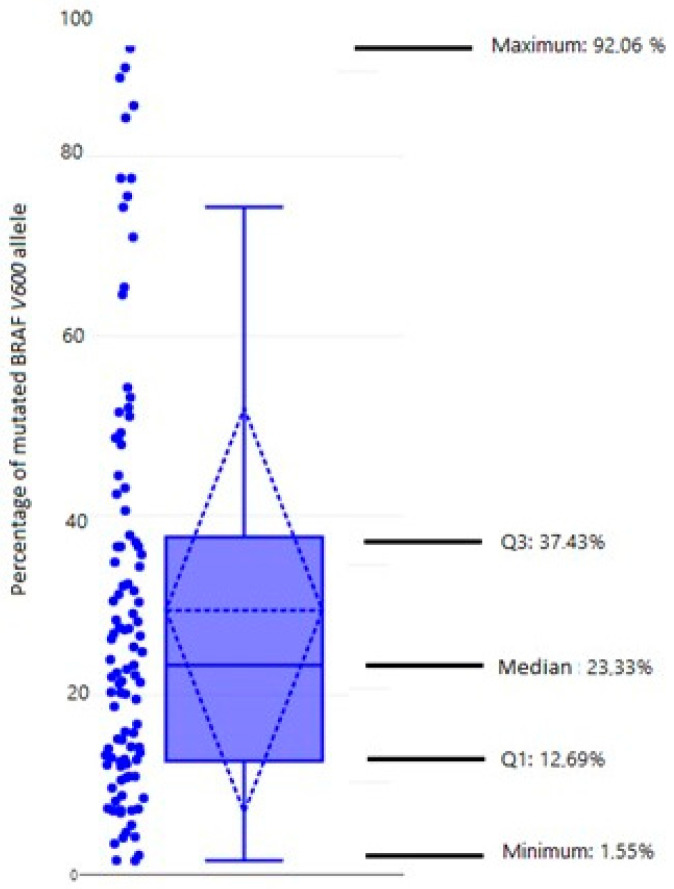
Box-plot showing the proportion of mutated allele in the primary tumor of all *BRAF* V600E positive patients. A single dot indicates the percentage of mutated allele for each individual *BRAF* positive patient. Marked on the picture: median, maximum, and minimum value, first and third quartile.

**Figure 2 biomedicines-12-00477-f002:**
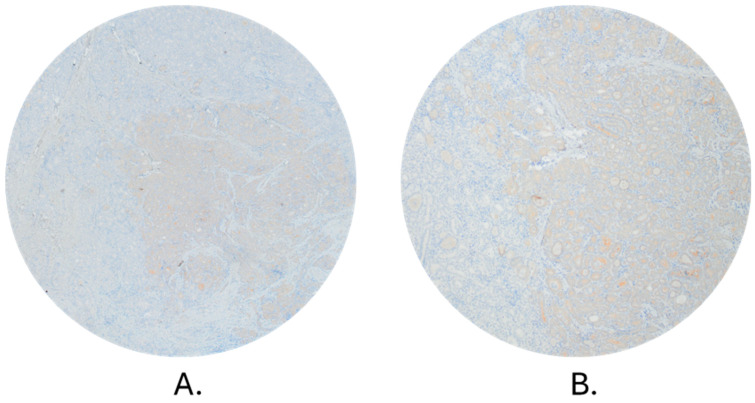
Staining of *BRAF* V600E mutated protein by IHC in tumors of different heterogeneity. Magnifications: (**A**) 40×; (**B**) 100×; (**C**) 200×; (**D**) 400×. CTRL Lymph node with metastatic melanoma showing positive immunohistochemical reaction to BRAF, positive control (100×).

**Table 1 biomedicines-12-00477-t001:** Sex and age of the patient, extent of the disease, and clinicopathological characteristics of the tumor.

Basic Characteristics of the Patient	All Patients		*BRAF* +	*BRAF* −	*p* Value
**Gender (n = 173); n (%)**					
**Women**	109 (63.0)		62	47	*p* = 0.257 ^a^
**Men**	64 (37.0)		42	22	
**Age, mean ± SD, years**	46.56 ± 16.62		49.58 ± 15.5	42.01 ± 17.3	*p* = 0.003 *^,b^
**Dissemination (n = 173); n (%)**					
**Confined to the thyroid**	81 (46.8)		50 (61.7)	31 (38.3)	*p* = 0.154 ^a^
**Loco-regional involvement**	61 (35.3)		40 (65.6)	21 (34.4)	
**Distant metastases**	31 (17.9)		14 (45.2)	17 (54.8)	
**Tumor size, mean ± SD, mm**	2.03 ± 1.33		1.84 ± 1.05	2.35 ± 1.60	*p* = 0.09 ^b^
**Perforation of the thyroid capsule (n = 167); n (%)**					
**YES**	37 (22.2)		21 (56.8)	16 (43.2)	*p* = 0.541 ^a^
**NO**	130(77.8)		81 (62.3)	49 (37.7)	
**Dissemination within the thyroid gland (n = 169); n (%)**					
**YES**	67(39.6)		38 (56.7)	29 (43.3)	*p* = 0.433 ^a^
**NO**	102(60.4)		64 (62.7)	38 (37.3)	
**Extrathyroidal spreading (n = 164); n (%)**					
**YES**	37	22.6	23(62.2)	14(37.8)	*p* = 0.866 ^a^
**NO**	127	77.4	77 (60.6)	50 (39.4)	
**Angioinvasion (n = 132); n (%)**					
**YES**	12 (9,1)		4(33.3)	8 (66.7)	*p* = 0.043 *^,a^
**NO**	120 (90.9)		76 (63.3)	44 (36.7)	
**Breakthrough of the lymph node capsule (n = 47); n (%)**					
**YES**	6	12.8	3	3	*p* = 0.309 ^a^
**NO**	41	87.2	29	12	

n = number of patients; ^a^: Chi^2^-test; ^b^: Mann–Whitney test; * statistically significant *p* value.

**Table 2 biomedicines-12-00477-t002:** Association of the frequency of the *BRAF* V600E mutated allele and dissemination status.

Normalized Percentage of *BRAF V600E* Allele				
**Disease dissemination**	Arithmetic mean (%)	CI− 95.00%	CI+ 95.00%	N
**Confined to the thyroid**	29.63	22.99	36.27	47
**Loco-regional involvement**	28.23	20.94	35.52	39
**Distant metastases**	33.61	21.44	45.78	14
**ANOVA test, F(2, 97) = 0.28287, *p* = 0.754**				

N = number of patients; χ^2^ = Chi^2^-test.

**Table 3 biomedicines-12-00477-t003:** Distribution of patients according to the frequency of the mutation allele *BRAF* V600E.

Number of Patients					
		Groups according to the normalized percentage of *BRAF V600E* allele			Total number
		Up to 40%	40% to 60%	More than 60%	
**Disease dissemination**	Confined to the thyroid	36 (74.5%)	7 (14.9%)	5 (10.6%)	47
	Loco-regional involvement	30 (76.9%)	5 (12.8%)	4 (10.3%)	39
	Distant metastases	11 (78.6%)	0 (0%)	3 (21.4%)	14
**Total**		76	12	12	100
**(χ^2^ (4, N = 100) = 3.257; *p* = 0.515)**					

N = number of patients; χ^2^ = Chi^2^-test; N = 100 including only *BRAF* V600E positive patients with data on normalized percentage of *BRAF* V600E allele.

## Data Availability

The data presented in this study are available on request from the corresponding author.

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
