# Peer review of "Association of BRAF V600E Mutant Allele Proportion with the Dissemination Stage of Papillary Thyroid Cancer"

_biomedicines, 2024, doi:10.3390/biomedicines12030477_

Round 1

Reviewer 1 Report

Comments and Suggestions for Authors

Potentially interesting, but needing the following corrections.

Introduction:

-          Thyroid cancer is a common tumor with a significant increase in incidence. Croatia ranks 4th according to the age-standardized incidence rate in Europe [1,2]. In recent years,  increased screening in the general population, primarily in developed countries, has led to an increase in the incidence of PTC, but without an accompanying increase in mortality 37 [3,4]. According to the 2017 SEER analysis (Lim et al. to be cited), the incidence-adjusted mortality for thyroid cancer is also increasing (please state this issue). Furthermore, it has been hypothesized (by many authors) that the incidence is at least in part independent from the empowered diagnostics….meaning that the evolving epidemiology reflects a real change in disease behaviour, likely related to environmental pressure.

-          When discussing about the genetics: a) briefly state that RAS mutations have not prognostic impact at all (reference: Significance of RAS Mutations in Thyroid Benign Nodules and Non Medullary Thyroid Cancer. Cancers (Basel). 2021 Jul 27;13(15):3785. doi: 10.3390/cancers13153785. PMID: 34359686; PMCID: PMC8345070.); b) address the relationship of the BRAF mutation with tumor vascularization highlighting the role of angiogenesis for DTC prognostication (references: 2007 BRAF mutation in papillary thyroid cancer: pathogenic role, molecular bases, and clinical implications. Endocrine Reviews 28 742–762. (doi:10.1210/er.2007-0007; Germline polymorphisms of the VEGF-pathway predict recurrence in non-advanced differentiated thyroid cancer. J Clin Endocrinol Metab. 2016 Nov 16:jc20162555. [Epub ahead of print] PubMed PMID: 27849428). This is even more important if considering that the only significant result was the relationship between BRAF presence/percentage and the angioinvasion; c) better address the studies reporting the correlation between BRAFV600E and clinical outcome (recurrence and mortality) in PTC (references: 2013 Association between BRAF V600E mutation and mortality in patients with papillary thyroid cancer. JAMA 309 1493–1501. (doi:10.1001/ jama.2013.3190; 2015 Association between BRAF V600E mutation and recurrence of papillary thyroid cancer. Journal of Clinical Oncology 33 42–50. (doi:10.1200/JCO.2014.56.8253)); d) also pose in evidence that the actual prognostic impact of the BRAF mutation alone is weak….the strong prognostic effect is actually related to the mutational duet BRAFV600E-TERT promoter mutations (reference: 2016 Prognostic effects of TERT promoter mutations are enhanced by coexistence with BRAF or RAS mutations and strengthen the risk prediction by the ATA or TNM staging system in differentiated thyroid cancer patients. Cancer 122 1370-1379 (doi:10.1002/cncr.29934); e) discuss the current role of the BRAF mutation in the ATA risk stratification for disease recurrence (reference: ATA guidelines on thyroid cancer 2016).

Methods:

-          Please define the angioinvasion.

-          I have some concerns about the way of determining the % of BRAF mutation. If this approach had been used in previous studies by the authors of different researchers, please state.

-          Please better address the issue of non-tumor cells infiltration as in the case of Hashimoto’s thyroiditis (reference: BRAF (V600E) assessment by pyrosequencing in fine needle aspirates of thyroid nodules with concurrent Hashimoto's thyroiditis is a reliable assay. Endocrine. 2014 Mar;45(2):249-55. doi: 10.1007/s12020-013-9994-y. Epub 2013 Jun 18.).

Results:

-          Please substitute dissemination with extension, differentiated confined to the thyroid, loco-regional involvement, distant metastases (meaning extracervical localization also including extracervical lymph nodes).

-          The normalized proportion of mutated BRAF V600E allele ranged from 1.55% to 92.06%. Considering the heterozygosity of the BRAF mutation, an ideal clonal PTC with 100% of tumor cells carrying the mutation should have a 50% of BRAFV600E allele percentage. How do you explain that 92.06%?

-          Try to clearly distinguish between clonal and non-clonal tumors.

-          The paper could be strikingly enriched by the analysis of clinical outcome, even though I think this is not feasible.

Discussion

Make a balance with the issues already raised concerning the Introduction.

Comments on the Quality of English Language

Minor editing

Reviewer 2 Report

Comments and Suggestions for Authors

This is very important work especially for the pathologists in practice. IHC for V600E , although many established successful performance in malignant melanoma, is often bewildering the pathologists who estimate the staining in the other organs such as GI tract cancers.

1. In addition to Fig 2 is it possible for the authors present more cases having heterogeneity and if there are inconsistent cases  for mutations.

2. Positive control cases of melanoma should be shown so that different profile is impressive for the readers.

3. Any arguments there, the mutation having no correlation with clinical performance can be categorized as "driver gene" in this context?

Round 2

Reviewer 1 Report

Comments and Suggestions for Authors

Accept as it is.

Comments on the Quality of English Language

.